# More effective drugs lead to harder selective sweeps in the evolution of drug resistance in HIV-1

Alison F Feder[1]*, Soo-Yon Rhee[2], Susan P Holmes[3], Robert W Shafer[2], Dmitri A Petrov[1†], Pleuni S Pennings[1,4†]

[1]Department of Biology, Stanford University, Stanford, United States; [2]Department of Medicine, Stanford University, Stanford, United States; [3]Department of Statistics, Stanford University, Stanford, United States; [4]Department of Biology, San Francisco State University, San Francisco, United States

**Abstract** In the early days of HIV treatment, drug resistance occurred rapidly and predictably in all patients, but under modern treatments, resistance arises slowly, if at all. The probability of resistance should be controlled by the rate of generation of resistance mutations. If many adaptive mutations arise simultaneously, then adaptation proceeds by soft selective sweeps in which multiple adaptive mutations spread concomitantly, but if adaptive mutations occur rarely in the population, then a single adaptive mutation should spread alone in a hard selective sweep. Here, we use 6717 HIV-1 consensus sequences from patients treated with first-line therapies between 1989 and 2013 to confirm that the transition from fast to slow evolution of drug resistance was indeed accompanied with the expected transition from soft to hard selective sweeps. This suggests more generally that evolution proceeds via hard sweeps if resistance is unlikely and via soft sweeps if it is likely.

*For correspondence: afeder@ stanford.edu

†These authors contributed equally to this work

**Competing interests:** The authors declare that no competing interests exist.

## Introduction

In the first two decades of the HIV epidemic, HIV became a prime example of fast evolutionary change, especially because of the evolution of drug resistance quickly after initiation of treatment. Nowadays, HIV treatments are more clinically effective and the evolution of drug resistance has become much slower and often does not occur for years if at all. The rate at which evolution occurs has been the subject of considerable recent interest in the evolutionary biology community. Although traditionally evolution was thought to be slow (*Darwin, 1859*), there are a growing number of examples of fast evolution to selective pressures such as pesticides (*Lopes et al., 2008*; *Daborn et al., 2001*; *Karasov et al., 2010*; *Palumbi, 2001*), industrialization (*Cook et al., 2012*), or antibiotics (*Laehnemann et al., 2014*; *Nair et al., 2007*). HIV represents an interesting case because its evolutionary speed in treated patients has changed drastically over time.

Population genetic theory suggests that whether populations evolve slowly or quickly is driven by the availability of adaptive mutations. In a large population with a high mutation rate, mutations may be available as standing genetic variation (pre-existing variation) or be generated anew every generation, allowing the population to adapt to its environment rapidly. If adaptive mutations are rare, because the population is small, the mutation rate is low, or only few specific mutations (or combinations of mutations) can help a population adapt, the population will likely adapt to its environment much more slowly.

The availability of adaptive mutations does not only change the rate of adaptation, it also changes how adaptation affects genetic diversity in a population. If adaptive mutations are rare, i.e.,

**eLife digest** In the early days of HIV therapy, the strains of the virus that infected patients frequently evolved drug resistance and the therapies would often eventually fail. These treatments generally involved using a single anti-viral drug. Nowadays, better therapies involving combinations of several anti-viral drugs are available and drug resistance in HIV is a much rarer occurrence. This means that now a particular therapy may be an effective treatment for an HIV-infected individual over much longer periods of time.

A theory of population genetics predicts that when it is easy for a population to acquire a beneficial genetic mutation – like one that provides drug resistance – multiple versions of that mutation may spread in the population at the same time. This is called a soft selective sweep. However, when beneficial mutations occur only rarely, it is expected that only one version of that mutation will take over in a population, which is known as a hard selective sweep.

Here, Feder et al. test this theory using data from 6717 patients with HIV who were treated between 1989 and 2013 using a variety of different drug therapies. The experiments aimed to find out whether the transition from the older drug therapies –where the virus frequently acquired resistance – to the newer, more effective drugs was associated with a transition from soft to hard sweeps.

Feder et al. find that HIV more often evolved drug resistance via soft sweeps in patients treated with the less effective drug combinations (like those given in the early days of HIV treatment), while hard sweeps were more common with the more effective drug combinations. This suggests that good drug combinations may allow fewer drug resistance mutations to occur in the HIV population within a patient. This may be because there are fewer virus particles in these patients, or because the specific combinations of mutations that provide resistance occur less often. Feder et al.'s findings are a step towards understanding why modern HIV treatments work so well, which will ultimately help us find better treatments for other infectious diseases.

less than one adaptive mutation occurs per generation in the population, the first successful mutation is likely to rise to high frequency before any subsequent adaptive mutations reach appreciable frequencies (see *Figure 1A*). This results in a hard selective sweep, in which the single adaptive mutation and the nearby linked mutations becomes fixed in the population (*Figure 1B*). Hard selective sweeps sharply reduce genetic diversity in the population (*Figure 1C*) (*Smith and Haigh, 1974*; *Kaplan et al., 1989*) in a similar manner to a strong genetic bottleneck.

In contrast, when adaptive mutations are common, i.e., more than one occurs per generation in the population, the same adaptive mutation may occur several times in a very short time span on different genetic backgrounds. These adaptive mutations can increase in frequency virtually simultaneously (*Figure 1D*) (*Pennings and Hermisson, 2006*) and multiple genetic backgrounds are therefore expected to reach substantial frequencies with no single genetic background dominating the population (*Figure 1E*). This pattern is known as a soft selective sweep and is expected to lead to almost no reduction of genetic diversity (*Figure 1F*) (*Pennings and Hermisson, 2006*), comparable to a mild bottleneck.

In HIV, the evolution of drug resistance was fast in patients on early anti-retroviral therapies (*Larder et al., 1989*), but current multi-drug regimens have substantially slowed the rate of evolution of resistance (*Martin et al., 2008*). Although clinically effective drugs have decreased the rate of emergence of drug resistance, it is not clear what effect that they have had on the evolutionary dynamics of within-patient HIV populations. Specifically, population genetic theory predicts that populations should evolve primarily by soft sweeps when resistance is likely and by hard sweeps when resistance is rare. In essence, soft sweeps should mark cases of resistance that arise deterministically through many origins while hard sweeps mark cases of rare, 'unlucky' resistance.

However, these predictions have not been tested in HIV or, in fact, in any natural population. In this study, we ask whether the transition from fast to slow evolution of drug resistance was indeed associated with a transition from soft to hard selective sweeps. If true, then in general, treatments in

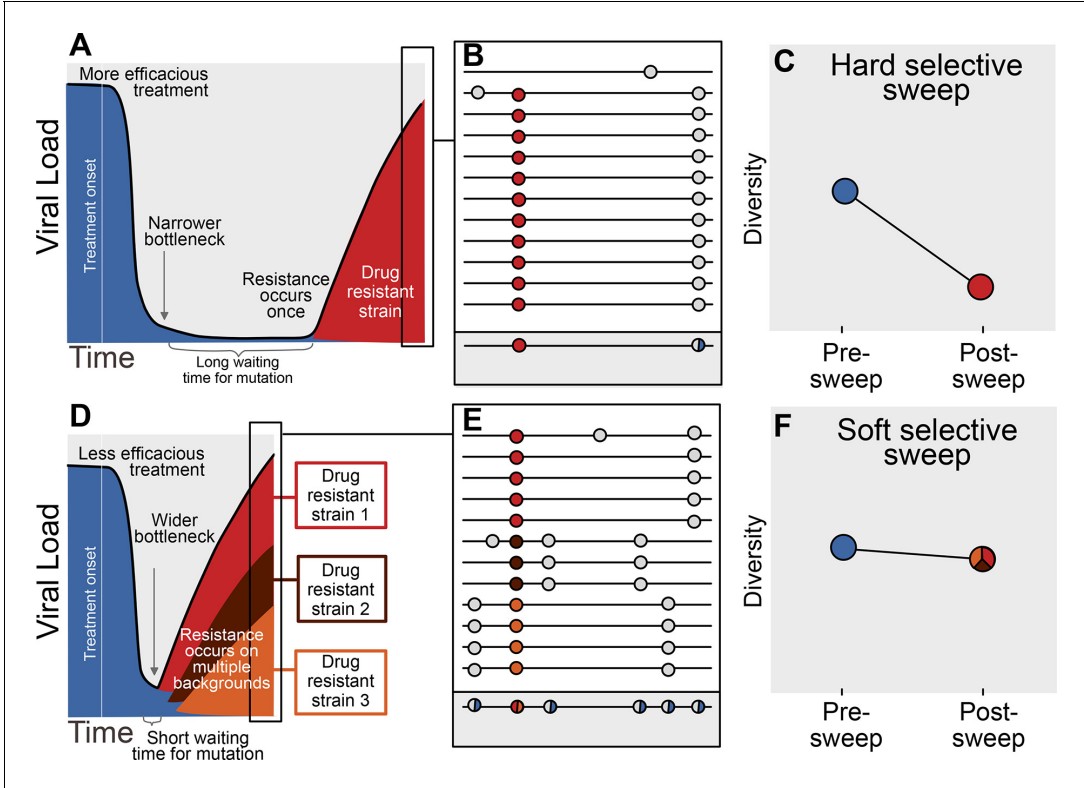

**Figure 1.** Prediction of drug resistance acquisition with more and less effective treatments. Among patients treated with more effective treatments (top), we predict HIV populations to have a lower probability of acquiring resistance per generation. As a result, the population must wait a long time for a beneficial genotype, so when resistance does occur, it will spread through the population in a hard selective sweep before other resistant genotypes emerge (**A**). Since resistance only occurs on a single genetic background (background mutations in grey), all sequences with resistance will be similar (**B**) and diversity following this type of selective sweep will be reduced (**C**). We can use the reduction of diversity to determine that a selective sweep is hard. In patients treated with less effective treatments (bottom), we predict HIV populations should have a higher probability of acquiring resistance per generation, so resistance will be acquired more quickly and selective sweeps of drug resistance mutations will be soft (**D**). We can detect these soft selective sweeps, because diversity remains high when resistance mutations on different genetic backgrounds rise in frequency simultaneously (**E,F**).

which resistance arises by soft sweeps might be predicted to have high rates of failure even before the rates of failure can be measured explicitly.

To test whether a transition from soft to hard sweeps has occurred, we look at the relationship between fixed drug resistance mutations (DRMs) and genetic diversity across 29 common anti-retroviral drug regimens. The expectation is that when hard selective sweeps predominate, we will find a negative correlation between the number of DRMs and genetic diversity in a population. On the other hand, when soft selective sweeps predominate, we expect to find no such correlation. We use 6717 HIV sequences from the same number of patients from the Stanford HIV Drug Resistance Database (*Rhee et al., 2003*, https://hivdb.stanford.edu/). These sequences contain information about the number of DRMs and, as we will further explain in the next paragraph, they also contain information about genetic diversity in the viral population.

Most sequencing of HIV populations in patients is done with the intent to discover DRMs for diagnostic and therapeutic reasons (*Dunn et al., 2011*). As such, in standard clinical practice, a sample from a patient's entire HIV population is amplified via PCR and then sequenced using the traditional Sanger method resulting in a single consensus sequence. Genetic diversity may result in ambiguous calls (also referred to as mixtures) in the reported sequence, so that a signal of within-patient genetic diversity is retained even though this sequencing approach generates only a single sequence per patient. We use the ambiguous calls to quantify within-patient genetic diversity (see *Figure 1B, E*, grey box), following several other studies (*Kouyos et al., 2011*; *Zheng et al., 2013*; *Li et al., 2012*;

*Poon et al., 2007*). Although ambiguous calls are an imperfect measure of diversity, it has been shown that the signal from ambiguous calls can be reproduced between laboratories (*Shafer et al., 2001*). By using ambiguous calls as a proxy for diversity, we are able to take advantage of a large number of HIV-1 sequences, allowing us to study the evolutionary dynamics of HIV drug resistance evolution in a historical perspective (*Rhee et al., 2003*).

Through examining HIV sequences of 6717 patients over the past two and a half decades in the presence of many different drug regimens, all sequenced using Sanger sequencing technology, we leverage ambiguous sequence calls to understand how the fixation of drug resistance mutations affects diversity. We find that, across all sequences, the presence of drug resistance mutations is associated with lower within-patient genetic diversity, marking the occurrence of selective sweeps. Second, we find that the extent of diversity reduction associated with drug resistance mutations varies with the clinical effectiveness of the treatment - effective drug treatments with low rates of virologic failure (such as NNRTI-based and boosted PI-based regimens) show strong reductions in diversity associated with each additional resistance mutation, a pattern more consistent with hard selective sweeps, whereas treatments that fail more often (such as regimens based only on NRTIs) show no reduction in diversity, a pattern consistent with soft selective sweeps. Although our results do not explain mechanistically how effective treatments lead to harder sweeps of drug resistance mutations, they suggest a more general principle: a lower rate of the production of adaptive mutations should be accompanied by harder sweeps.

## Results

### Sequences and patients

We collected sequences of reverse transcriptase and/or protease genes from 6717 patients from the Stanford HIV Drug Resistance Database (*Rhee et al., 2003*). The sequences come from 120 different studies that were performed between 1989 and 2013. The 6717 patients represent all individuals in the database who were treated with exactly one drug regimen, usually comprising a combination therapy of multiple drugs (see Materials and methods: Data collection & filtering). The patients' viral populations were sequenced after at least some period of treatment, although treatment may or may not have ceased at the time of sequencing and treatment may or may not have failed. This virus from a patient was amplified via PCR, sequenced using the Sanger method and then was reported to the database as a single nucleotide sequence. We call this dataset the D-PCR dataset, for direct PCR.

All 6717 patients received some type of therapy (between 1 and 4 drugs), with the majority (77%) receiving a regimen of three drugs. Nearly all patients received one or two nucleoside reverse transcriptase inhibitors (NRTI), usually paired with a non-nucleoside reverse transcriptase inhibitor (NNRTI) or a protease inhibitor (PI), which was boosted with a low dose of ritonavir in some patients. HIV subtypes were varied, with the majority being B (36%), C (34%) or CRF01_AE (13%). None of the remaining subtypes contributed more than 5% of the total sample.

An additional dataset, which we call the clonal dataset, consisted of 11,653 sequences from 740 patients with multiple sequences per patient isolated through clonal amplification and subsequent Sanger sequencing. The clonal dataset was used for validation purposes only.

### Ambiguous calls are a good proxy for genetic diversity

We are interested in the effect of drug resistance evolution on within-patient genetic diversity, but in our main dataset, we only have one sequence per patient. To use this large dataset for our purposes, we therefore use ambiguous nucleotide calls as a proxy for within-patient genetic diversity. Although results from previous studies suggest that this approach is valid (*Kouyos et al., 2011*; *Shafer et al., 2001*), we independently validate this measure through comparing the D-PCR and clonal datasets. Using the clonal sequences, within-patient diversity ($\pi$) can be computed directly, giving an estimate of genetic diversity per site that does not rely on ambiguous calls. We compared the proportion of sequences with ambiguous calls at a site in the D-PCR dataset to the within-patient diversity ($\pi$) at that site in the clonal dataset (see Materials and methods: Validation of ambiguous calls as genetic diversity measure). We find that clonal within-host nucleotide diversity has a high positive correlation with the percentage of nucleotide calls ambiguous in the D-PCR dataset ($r$ =

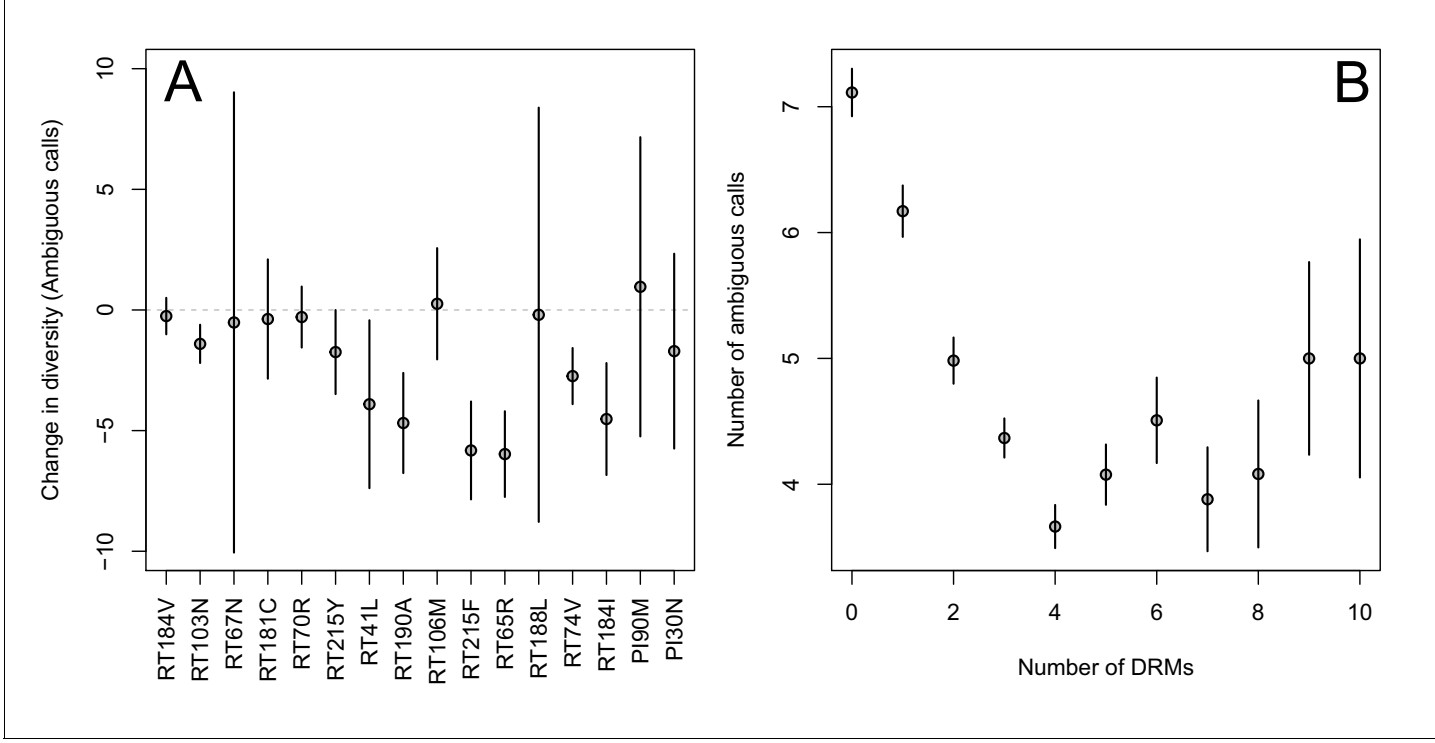

**Figure 2.** Effect of DRMs on sequence diversity. (**A**) For the most common reverse transcriptase and protease mutations, 95% confidence intervals are drawn for the difference in diversity associated with a single derived mutation. For each DRM, the mean diversity among patients with a fixed ancestral state at the focal locus is compared to those patients with that fixed non-ambiguous DRM. All sequences have no additional DRMs. All DRMs occurring at least 5 times with these specifications are included. (**B**) The effect of multiple DRMs on diversity is shown as the average diversity level of sequences decreases conditional on number of fixed drug resistance mutations present. Means ± SE are plotted among all patients in the D-PCR dataset.

The following figure supplements are available for figure 2:

**Figure supplement 1.** Diversity and the number of drug resistance mutations by treatment categories.

**Figure supplement 2.** Effect of multiple DRMs on sequence diversity separated by subtype.

0.91, p<2.2 × $10^{-16}$). A similar pattern holds at the amino acid level ($r$ = 0.85, p<2.2 × $10^{-16}$). We therefore conclude that ambiguous calls are a reliable proxy for within-patient genetic diversity.

## Drug resistance mutations (DRMs) lower within-patient diversity

We ask whether across all sequences, the presence of a drug resistance mutation (DRM) is associated with lower within-patient diversity, the classical signature of a selective sweep (*Smith and Haigh, 1974*). For each sequence, we therefore count the number of DRMs present that are relevant for the treatment the patient was taking (*Bennett et al., 2009*) (i.e., a mutation that confers resistance to a particular class of drugs was counted as a DRM only if the patient was actually being treated with that drug; see Materials and methods: Sequence processing for more information).

First, for the most common reverse transcriptase and protease DRMs, we compare sequences that have exactly one DRM with sequences that have zero DRMs (i.e., ancestral state at all possible DRM sites). We plot the difference in within-patient diversity between the two groups with 95% confidence intervals in *Figure 2A*. Among reverse transcriptase and protease DRMs, sequences with the DRM have lower diversity than those with the ancestral state in 14 of 16 cases, with 7 of the 16 being significantly lower at the 95% confidence level. This reduction in diversity is consistent with expectations after a selective sweep (*Smith and Haigh, 1974*).

Second, we looked at the effect of multiple DRMs on within-patient diversity, as we hypothesize that multiple fixed DRMs may decrease diversity even more than a single DRM. This could result

from sequential selective sweeps of single DRMs each reducing diversity, or from a single selective sweep that fixes multiple DRMs. Indeed, we find that for sequences that have between 0 and 4 DRMs, additional DRMs are associated with reduced genetic diversity (*Figure 2B*, p-value for *t*-test between diversity among sequences with 0 versus 1 is $7.2 \times 10^{-4}$, between 1 and 2 is $1.6 \times 10^{-5}$, between 2 and 3 is $1.1 \times 10^{-2}$, between 3 and 4 is $2.5 \times 10^{-3}$). After 4 DRMs, subsequent DRMs do not significantly reduce diversity further. The observed pattern of DRMs associated with reduced diversity is mainly driven by the patients receiving NNRTI or boosted PI-based treatments, as can be seen when separating the above analysis by drug treatment category (*Figure 2—figure supplement 1C,D*). Among patients treated with NRTIs alone or with unboosted PIs, this pattern is much less clear (*Figure 2—figure supplement 1A,B*). The observed pattern holds across the each of the most common subtypes separately (*Figure 2—figure supplement 2*).

## Clinical effectiveness of anti-retroviral regimens

We have now shown that in general, each additional DRM is associated with reduced diversity, which is consistent with expectations of selective sweeps. We want to assess how this effect depends on the clinical effectiveness of the treatment. For the most common drugs in our dataset, we assess clinical drug treatment effectiveness categorically and quantitatively.

As a categorical approach, we separated regimens based on general clinical HIV-treatment recommendations where NNRTI-based treatments are preferred to NRTI-based treatments, and treatments based on ritonavir-boosted PIs (PI/r) are preferred to treatments based on unboosted PIs. These more and less effective groupings are the basis of comparisons in our parametric approach described below.

To measure effectiveness quantitatively, we conducted a literature search to determine the percentage of patients who have remained virologically suppressed after one year of treatment (see Materials and methods: Clinical effectiveness of antiretroviral treatments, *Supplementary files 1– 3*) for 21 different treatments with at least 50 sequences per treatment in our D-PCR dataset (see description of abundant treatment dataset in Materials and methods: Data collection & filtering for more information.) This quantitative measure was used as the basis for our non-parametric approach described below.

The two measures (categorical and quantitative) correspond well and clinical treatment effectiveness ranged widely, from very low effectiveness (5% of patients virologically suppressed after one year of treatment on AZT monotherapy) to very high effectiveness (100% of patients virologically suppressed after one year of treatment on 3TC+AZT+LPV/r) (*Figure 3A,B*).

## High treatment effectiveness associated with stronger diversity reduction

We hypothesize that effective treatments (such as those containing an NNRTI or boosted PI) likely make adaptation in viral populations limited by the generation of mutations and these populations should thus experience harder selective sweeps leading to a sharp reduction in diversity accompanying each additional DRM. Less effective treatments on the other hand (such as those containing only NRTIs or unboosted PIs) likely allow replication of fairly large HIV populations so that adaptation is not limited by the generation of mutations. They should thus experience soft selective sweeps and little or no reduction of diversity with each additional DRM. Below we test this hypothesis by assessing the reduction of diversity associated with the presence of a DRM among treatments that vary in clinical effectiveness.

Before we are able to test this hypothesis, we have to deal with a peculiarity of our data. We found that even for sequences that carried no resistance mutations, more ambiguous calls were reported over time. This is likely due to increased awareness of genetic diversity in the HIV community, and not because diversity actually increased. We therefore employed a p-thinning routine to repeatedly subsample the data so that diversity measures would be comparable across years (see Materials and methods: p-thinning to adjust for the effect of year).

To determine whether the effect of DRMs on within-patient diversity depends on clinical treatment effectiveness, we first fit a generalized linear mixed model (GLMM) using the number of DRMs and sequence length to predict diversity as measured by the number of ambiguous calls. Because we found the subsampled number of ambiguous calls to be distributed according to a negative

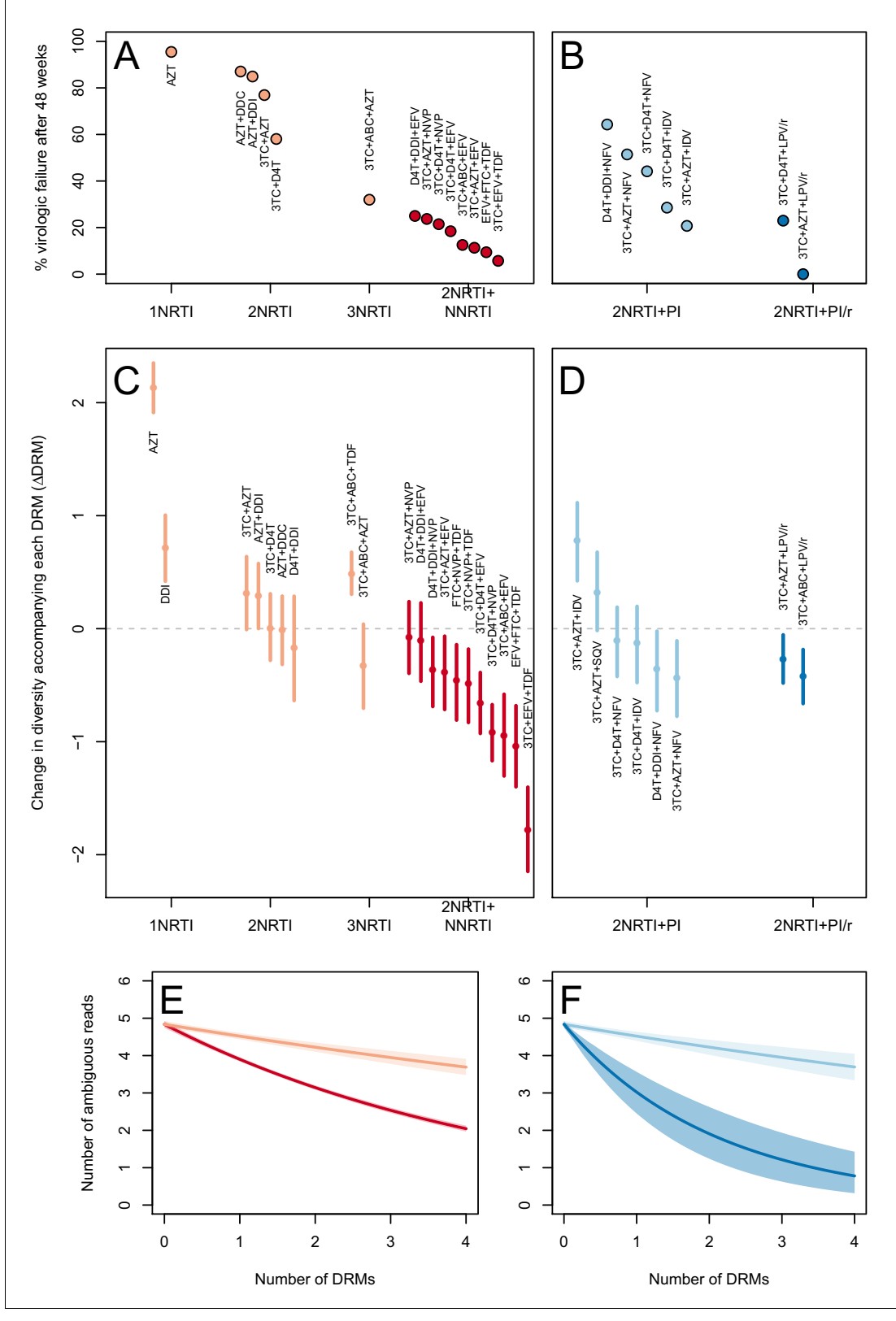

**Figure 3.** Drug resistance mutations are correlated with diversity reduction differently in different types of treatments. Treatment effectiveness from literature review (percentage of patients with virologic suppression after ~48 weeks) showed positive correspondence with clinical recommendation among RTI regimens (**A**) and PI+RTI regimens (**B**). Diversity reduction accompanying a DRM ($\Delta_{DRM}$ from the GLMM) lower among the more effective

*Figure 3 continued on next page*

*Figure 3 continued*

and clinically recommended treatments among RTI treatments (**C**) and RTI+PI treatments (**D**). 95% confidence intervals are plotted by excluding the highest 2.5% and lowest 2.5% of GLMM random effect fits of the 1000 subsampled datasets and treatments are ordered by mean $\Delta_{DRM}$ within treatment categories. Generalized linear model fits show significantly different slopes for NNRTI treatments versus NRTI treatments (**E**) and PI/r treatments versus PI treatments (**F**). Confidence intervals are plotted by excluding the highest 2.5% and lowest 2.5% of GLM fits to 1000 subsampled datasets.

The following figure supplements are available for figure 3:

**Figure supplement 1.** Drug resistance mutations are correlated with diversity reduction differently in different types of treatments when all years are included.

**Figure supplement 2.** Drug resistance mutations are correlated with diversity reduction differently in different types of treatments with un-truncated data.

binomial distribution, we used a negative binomial error distribution (see Materials and Methods: Quantifying the relationship between clinical effectiveness and diversity reduction). For each treatment, we report the total effect of the number of DRMs on diversity ($\Delta_{DRM}$) as the sum of the treatment-specific random effect plus the overall fixed effect from the GLMM ($\Delta_{DRM,t} + \Delta_{DRM,all}$). These $\Delta_{DRM}$ coefficients are plotted in *Figure 3C,D*.

We included only treatments that had at least 15 sequences and a sufficient number of observed patients with different numbers of DRMs (see description of abundant treatment dataset in Materials and methods: Data collection & filtering for more information). We also excluded 45 sequences sampled before 1995 which had an extreme influence on the p-thinning routine (see Materials and methods: p-thinning to adjust for the effect of year for details). The analysis that includes these sequences leads to qualitatively similar results (see the supplement). We also used only sequences with at most 4 DRMs. This captures the initial change in diversity due to the fixation of DRMs, and allows the $\Delta_{DRM}$ measure not to be driven by few patients with many DRMs. The same analysis with all sequences is repeated in the supplement and yields qualitatively similar results. Lower $\Delta_{DRM}$ values correspond to a bigger decrease in diversity associated with each DRM - a pattern more consistent with hard selective sweeps.

We find that most of our $\Delta_{DRM}$ estimates are qualitatively consistent with expectations: effective treatments have lower $\Delta_{DRM}$ values than less effective treatments. Most NNRTI-based treatments are associated with a reduction of diversity per DRM. In 9 of 11 NNRTI-based treatment regimens, $\Delta_{DRM}$ is significantly below 0 (*Figure 3C*). This pattern suggests the presence of hard sweeps, although there is variation in effect size. Less effective treatments containing only NRTIs were generally associated with a smaller or no reduction in diversity per DRM. In some cases, such as DDI or AZT

**Table 1.** Model fits for the fixed effects from GLMMs fit to subsampled data. See Materials and methods: Quantifying the relationship between clinical effectiveness and diversity reduction for further explanations of coefficients. Means of model fits for 1000 independent subsamples are reported for the three different subsampling and model-fitting regimes. 95% confidence intervals (excluding the top 2.5% and bottom 2.5% of coefficent fits) are given in parentheses.

| | $\alpha_{all}$ (Intercept) | $\Delta$ (Number of DRMs) | $\gamma$ (Length) |
|---|---|---|---|
| **1995+, $\leq$ 4 DRMs** | -0.78 | -0.16 | 0.0030 |
| | (-0.91,-0.65) | (-0.17,-0.14) | (0.0029,0.0032) |
| **1995+, All DRMs** | -0.89 | -0.097 | 0.0030 |
| | (-1,-0.77) | (-0.11,-0.088) | (0.0029,0.0032) |
| **1989+, $\leq$ 4 DRMs** | -1.20 | -0.15 | 0.0030 |
| | (-1.4,-1.1) | (-0.17,-0.14) | (0.0028,0.0032) |

monotherapy, there was even an increase of diversity associated with DRMs (a significantly positive $\Delta_{DRM}$ value, see *Figure 3C*). This pattern is suggestive of soft sweeps.

We found a slightly positive value of $\Delta_{DRM}$ for the NRTI-based regimen 3TC+ABC+TDF, a treatment which is known to often lead to rapid treatment failure (*Gallant et al., 2005*; *Khanlou et al., 2005*). Among the most negative $\Delta_{DRM}$ values of the NNRTI-based treatments were the treatments 3TC+ABC+EFV and EFV+FTC+TDF, which have long been on the list of recommended treatments in the USA (*Department of Health and Human Services, 2015*) (until their recent replacement with INSTI-based treatments which are not in our dataset).

Among treatments containing PIs, both of the effective boosted-PI treatments had $\Delta_{DRM}$ significantly below 0 (*Figure 3D*). The less effective unboosted PI treatments had $\Delta_{DRM}$ values on average closer to 0, and two of five unboosted treatments had a $\Delta_{DRM}$ value above 0. As expected, we find that the LPV/r treatment in our dataset has a much lower $\Delta_{DRM}$ value than the NFV treatments, consistent with the recommendations that LPV/r is preferable to NFV in relation to drug resistance prevention (*Walmsley et al., 2002*).

When the analysis is done without truncating the number of DRMs (*Figure 3—figure supplement 1*), results are qualitatively similar, but the values of specific treatments have shifted due to the effects of sequences with varying numbers of DRMs. When the 45 sequences before 1995 are included (*Figure 3—figure supplement 2*), results are also qualitatively similar. The fixed effects from the three different GLMM fits can be found in *Table 1*.

To quantify and further test the observation that more clinically effective treatments lead to the greater diversity reduction per fixed DRM, we use two primary approaches, one parametric and the other non-parametric.

## Parametric approach

We create 1000 datasets with p-thinned nucleotide calls and fit generalized linear models (GLMs) with negative binomial error distributions to each of the 1000 datasets. For each subsample, we separately fit the effect of increased numbers of DRMs on diversity for inferior and superior RTI treatments (1, 2, or 3 NRTIs and 2NRTIs+1NNRTI, respectively) and inferior and superior PI-based treatments (2NRTIs + PI and 2NRTIs + PI boosted with ritonavir).

We first use this model to assess the difference between inferior and superior RTI treatments (1,2,3 NRTIs and 2NRTIs+1NNRTI). We find that among sequences from patients receiving highly effective treatments with NNRTIs each fixed DRM is associated with a mean additional 13.6% reduction in diversity as compared to populations with no DRMs (95% confidence interval, 13.0%–14.3%). Among sequences from patients receiving less effective treatments with only NRTIs, each fixed DRM is associated with a mean additional 4.3% reduction in diversity compared to populations with no DRMs (95% confidence interval, 3.2%–5.3%) (*Table 2*). The relative effect of DRMs on diversity for more and less effective treatments can be seen in *Figure 3E*, where the dark red line is the mean model fit for effective NNRTI treatments and the light red line is the mean model fit for less effective NRTI treatments for a fixed sequence length (800 nucleotides).

**Table 2.** Coefficients from GLMs fit to subsampled data. See Materials and methods: Quantifying the relationship between clinical effectiveness and diversity reduction for further explanations of coefficient descriptions. Means for 1000 independent subsamples are reported for the three different subsampling and model-fitting regimes. 95% confidence intervals (excluding the top 2.5% and bottom 2.5% of coeffcient fits) are given in parentheses.

| | $\alpha_{all}$ **(Intercept)** | $\gamma$ **(Length)** | $\Delta_{1,2,3NRTI}$ | $\Delta_{2NRTI+NNRTI}$ | $\Delta_{2NRTI+PI/r}$ | $\Delta_{2NRTI+PI}$ |
|---|---|---|---|---|---|---|
| **1995+, $\leq$ 4 DRMs** | -0.46 | 0.0025 | -0.068 | -0.22 | -0.067 | -0.48 |
| | (-0.58,-0.35) | (0.0024,0.0027) | (-0.085,-0.050) | (-0.23,-0.20) | (-0.094,-0.043) | (-0.68,-0.30) |
| **1995+, All DRMs** | -0.62 | 0.0026 | -0.049 | -0.13 | -0.0035 | -0.20 |
| | (-0.72,-0.51) | (0.0025,0.0028) | (-0.061,-0.036) | (-0.14,-0.12) | (-0.019,0.011) | (0.30,-0.12) |
| **1989+, $\leq$ 4 DRMs** | -0.91 | 0.0026 | -0.063 | -0.22 | -0.065 | -0.48 |
| | (-1.00,-0.78) | (0.0024,0.0027) | (-0.084,-0.044) | (-0.23,-0.20) | (-0.098,-0.034) | (-0.76,-0.27) |

To assess the difference in effect of DRMs on diversity between NRTIs and NNRTIs, we performed a matched Wilcoxon sign-rank test comparing the slopes of the model fits to the 1000 subsampled datasets. We found treatments containing NNRTIs had significantly lower slopes than those treatments containing only NRTIs ($p < 2.2 \times 10^{-16}$). In fact, in every one of the 1000 subsampled datasets, the NNRTI category had a more negative slope than the NRTI category.

Among PI-based treatments, we found that sequences from effective treatments based on boosted PIs showed a mean additional reduction in diversity of 30.3% with each fixed DRM as compared to populations with no DRMs (95% confidence interval, 19.0%–42.3%). In contrast, sequences from less effective treatments based on unboosted PIs showed a smaller decrease in diversity of 4.3% (95% confidence interval, 2.7%-6.0%) with each fixed DRM (*Table 2*). The effect of DRMs on

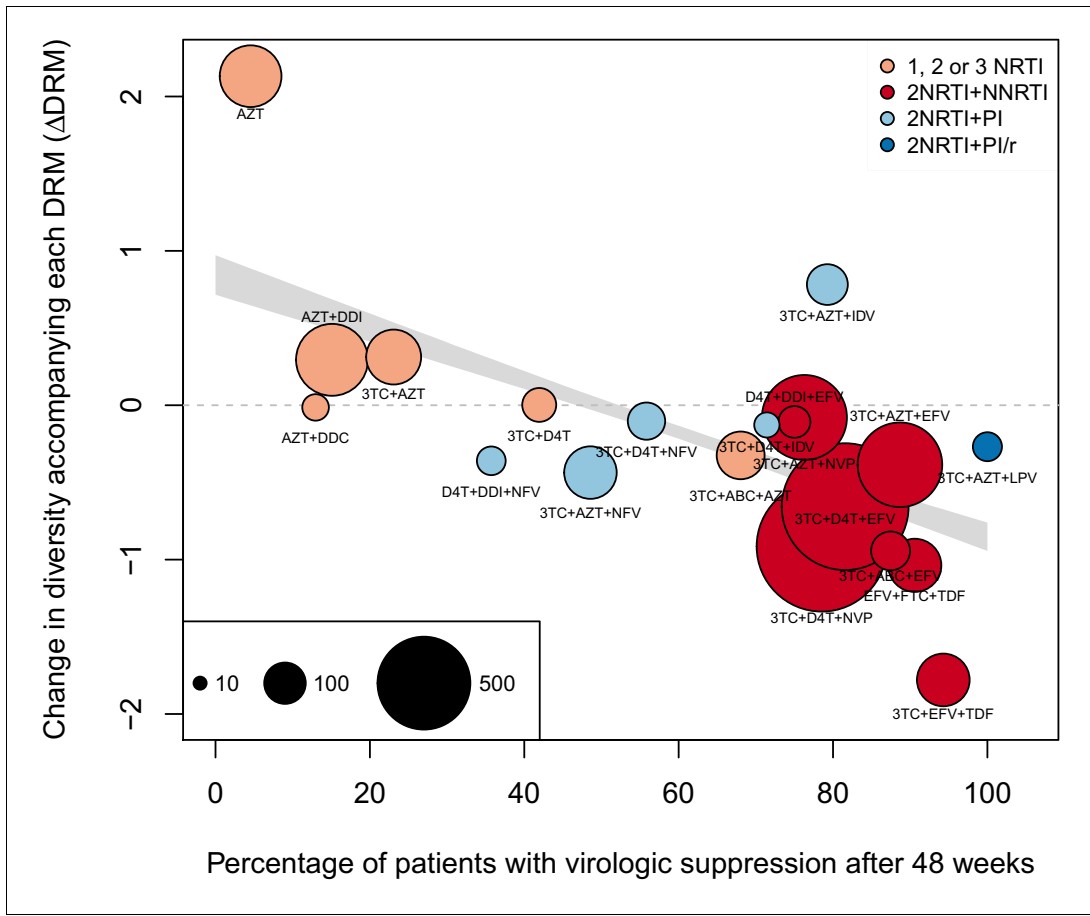

**Figure 4.** Nonparametric test shows negative correlation between treatment effectiveness and $\Delta_{DRM}$. Negative relationship between treatment effectiveness (percentage of patients with virologic suppression after ~48 weeks) and $\Delta_{DRM}$ as fit within the GLMM among all treatments for which we have recorded effectiveness information. Treatments included are plotted at the mean $\Delta_{DRM}$ from GLMMs fit to 1000 iterations of data subsampling. Points for each treatment are sized in proportion to the number of sequences from patients given that treatment (see legend in the lower left), and are colored based on the treatment type (see legend in the upper right). The black line shows the mean effect of treatment effectiveness on $\Delta_{DRM}$ from linear regressions fits to the 1000 subsampled datasets and the grey band shows a 95% confidence interval (excluding the top and bottom 2.5% of fits).

The following figure supplements are available for figure 4:

**Figure supplement 1.** Nonparametric test shows negative correlation between treatment effectiveness and $\Delta_{DRM}$ when 45 sequences from before 1995 are included.

**Figure supplement 2.** Nonparametric test shows negative correlation between treatment effectiveness and $\Delta_{DRM}$ when sequences with any number of DRMs are included.

diversity among patients treated with boosted PIs is significantly lower than among patients treated with unboosted PIs (paired Wilcoxon rank-sign test, p<2.2 × $10^{-16}$). The relative effect of these $\Delta_{DRM}$ coefficients can be seen in *Figure 3F*, where the dark blue line is the mean model fit for effective PI/r treatments and the light blue line is the mean model fit for less effective unboosted PI treatments, both for a sequence of length 800 nucleotides.

In both treatment classes (RTI-based and PI-based), we find that the fixation of a DRM under more effective treatments leads to a much stronger reduction of diversity than the fixation of a DRM under less effective treatments.

## Non-parametric approach

As a second, non-parametric approach, we did a test with the ability to compare the effects of all treatments (RTIs and PIs) directly. In order to do this, we fit a linear regression among the 20 treatments for which we had information about effectiveness to predict the reduction of diversity per DRM (taken as the random effects fit by treatment in the GLMM above) from treatment effectiveness. We observe a negative relationship between $\Delta_{DRM}$ and treatment effectiveness (*Figure 4*). We find that a 10% increase in treatment effectiveness is associated with 0.17 fewer ambiguous nucleotide calls with each DRM (95% confidence interval across 1000 fits, [-0.15, -0.19]). This means that patients given treatments with 50% effectiveness have approximately the same amount of diversity whether they have 0 or 3 DRMs, but patients on treatments with 80% effectiveness have 54% fewer if they have 3 DRMs as compared to 0. This is a substantial decrease in diversity.

Within this analysis, we find that among regimens with >30% effectiveness, those with the highest $\Delta_{DRM}$ values were all unboosted PI treatments (light blue points, *Figure 4*). Although the effectiveness value for these treatments is high by our metric, it has been observed that very high compliance rates are necessary for virologic suppression through unboosted PIs relative to NNRTIs and boosted PIs (*Shuter, 2008*). This is likely because the half-life of most unboosted PIs is sufficiently short that missed doses create conditions in which the virus is not suppressed. It is therefore possible that resistance is primarily found in patients with poor adherence, and in these patients, resistance could arise deterministically via soft sweeps. In addition, we may underestimate the number of DRMs for PIs, as some DRMs are likely to be outside of the sequenced regions (*Juno et al., 2012*; *Rabi et al., 2013*). These undetected DRMs would serve to weaken the signal for PI treatments compared to RTI treatments.

We find a similar negative relationship between $\Delta_{DRM}$ and treatment effectiveness when including 45 sequences before 1995 (*Figure 4—figure supplement 1*) and including sequences with more than 4 DRMs (*Figure 4—figure supplement 2*).

## Discussion

Treatment for HIV-1 represents an enormous success of modern medicine. Whereas early antiretroviral treatments were associated with fast evolution of drug resistance and high rates of treatment failure, there are currently many combinations of drugs that are successful at keeping HIV-1 at low or undetectable levels for many years, preventing the evolution of resistance and the progression to AIDS. Indeed, the evolution of drug resistance has become fairly uncommon (*Lee et al., 2014*). In this paper, we have shown that this shift from fast to slow evolution of drug resistance has been accompanied by a corresponding shift from soft selective sweeps (in which the same DRM occurs on multiple genetic backgrounds) to hard selective sweeps (in which a DRM occurs only a single time). This suggests that modern treatments have brought HIV into a regime where the viral population must wait until the correct mutation or combination of mutations is generated. This also means that for any given patient the acquisition of drug resistance has become at least partly an unlucky occurrence, in sharp contrast to the early days of HIV treatment in which all patients predictably failed treatment. Harder sweeps within well-treated patients are also consistent with the overall decrease in the rate of resistance.

We want to study how the evolutionary dynamics of selective sweeps have changed in the evolution of drug resistance over the past two and a half decades. Because it would be unethical to give subpar treatment to HIV infected patients, we can only investigate this question using historical data. The only type of data that is available for a wide range of treatments and time points is Sanger sequencing data, due to its importance in HIV research and diagnostic testing. Although we often

have only one sequence per patient, it is important to note that there is still information present in these sequences about genetic diversity and selective sweeps. First, we used ambiguous nucleotide calls as a proxy for genetic diversity. Although we are not the first study to do so (*Kouyos et al., 2011*; *Li et al., 2012*; *Poon et al., 2007*; *Zheng et al., 2013*), as far as we are aware, we are the first to use clonal sequences to validate its accuracy as a measure. Second, we can use the number of fixed drug resistance mutations to determine how much adaptation to treatment has taken place. We used a fairly conservative list of DRMs that are very unlikely to fix in the absence of the drug (*Mesplède et al., 2013*; *Gonzalez et al., 2004*; *Cong et al., 2007*). Therefore, DRMs must have fixed as a result of strong positive selective pressure imposed by the drug, and are indicative of recent selective sweeps. We can then look at the correlation between the number of drug resistance mutations and genetic diversity and see if that relationship has changed across treatments and time. Because this approach relies only on widely-available Sanger sequences, we were able to compare 29 different treatments, from AZT monotherapy to treatments based on boosted PIs, sampled across more than two decades (1989–2013).

Examining the relationship between DRMs and diversity recapitulates expected results: we first find that across the entire dataset, sequences with a single DRM have lower genetic diversity than sequences without any DRMs. This result confirms a finding from a previous much smaller study looking at patients on NNRTI based treatments (*Pennings et al., 2014*). In addition, we find that having more DRMs is associated with a greater reduction in diversity. This pattern could be generated by successive selective sweeps, in which DRMs are fixed one by one and each selective sweep lowers the diversity further. Alternatively, multiple DRMs may have fixed simultaneously in a single selective sweep.

The key result of this paper (illustrated in *Figure 1*) is that drug resistance mutations are associated with reduced diversity in patients on effective treatments, whereas this pattern is not seen among patients on older treatments with low clinical effectiveness. For example, among patients given treatments with 50% effectiveness, sequences with 3 DRMs are predicted to have approximately the same number of ambiguous calls as those with 0 DRMs. In contrast, among those patients given treatments with 80% effectiveness, sequences with 3 DRMs are predicted to have over 50% fewer ambiguous calls than those with 0 DRMs, a substantial decrease in genetic diversity. Thus, the higher the treatment effectiveness, the more DRMs are associated with low genetic diversity. This is consistent with drug resistance evolution dominated by soft selective sweeps when failure rates were high, transitioning over time to evolution dominated by hard selective sweeps as treatments improved and failure rates became much lower. Clinically effective treatments are thus characterized by a more frequent occurrence of hard selective sweeps.

It is of interest to compare our new results to a previous study by one of us (*Pennings et al., 2014*). Sequences in that study came from patients who were mostly treated with EFV + IDV, a combination that never became common and is not represented here, but which has an estimated effectiveness of 75% (*Staszewski et al., 1999*). The study examined selective sweeps in those patients by looking at the fixation of a particular DRM, at amino acid 103 in RT, which changes from Lysine (K, wild type) to Asparagine (N, resistant). K103N is special because it can be caused by two different mutations, as the wild type codon AAA can mutate to AAT or AAC, both of which encode Asparagine. When focusing on patients whose virus acquired the K103N mutation, the study found that in some patients, both the AAT and AAC codons were found (which is clear evidence that a soft sweep has happened, see Figure 1 in the original paper), whereas in other cases only one of the two was there (which suggests that a hard sweep may have happened, see Figure 2 in the original paper). Because of the detailed data available for these patients, it was shown that both soft and hard sweeps were occurring almost equally often. Placing the 75% effectiveness of EFV+IDV in the context of our above results, this is also what we would have predicted. Now we know that this result (hard and soft selective sweeps occur) is not necessarily something that will be generally true for HIV, but rather it is a function of the effectiveness of the treatment. Had Pennings et al. had data from a much worse or much better treatment, they might have concluded that hard sweeps or soft sweeps were the rule in HIV.

The transition to highly effective treatments and hard selective sweeps was not abrupt. As visible in *Figure 3*, treatment effectiveness and $\Delta_{DRM}$ do not cluster into distinct groups based solely on the number and type of component drugs. The incremental changes in effectiveness and the evolutionary dynamics are worth noting because a simplified narrative sometimes suggests that solving the

drug resistance problem in HIV was achieved simply by using three drugs instead of two (*Stearns and Koella, 2007*). According to this narrative, HIV can always easily evolve resistance when treatment is with one or two drugs, but it is virtually impossible for the virus to become resistant to three drugs. In truth, only some specific combinations seem to lead to more favorable evolutionary dynamics for patients (as seen in *Figure 3A,B*).

Several potential mechanisms could drive the observation that more effective drug combinations drive hard sweeps in within-patient populations of HIV-1. Better drugs may allow for a faster collapse of population size, decreasing the probability that one or more 'escape' DRMs occurs ([*Alexander et al., 2014*], although see [*Moreno-Gamez et al., 2015*]). Alternatively, suppressed HIV populations may continue replicating at small numbers, and better drugs may cause this replicating population to be smaller than among patients given inferior drugs. Similarly, a treated patient may retain a reservoir of HIV unreachable by the treatment, and better drugs may make this reservoir smaller. If newer drugs have fewer side effects and therefore improve adherence among patients, this too could result in a smaller within-patient population size among patients treated with better drugs, and contribute to the decreased production of resistant genotypes. The effect could also be driven by standing genetic variation, if worse drugs would allow pre-existing mutations to establish, whereas better drugs make this less likely, for example, if a pre-existing mutation could only establish if it occurred in a small and specific compartment (*Moreno-Gamez et al., 2015*). Finally, more effective drug combinations may simply require more mutations on the same background (i.e., a higher genetic barrier to resistance [*Tang and Shafer, 2012*]), so the effective mutation rate to a fully resistant genotype is lower. Our data do not give us sufficient resolution to distinguish between these hypotheses, and the true dynamics may be a combination of all of these factors. However, all these potential mechanisms work to reduce the rate of production of resistant genotypes and the principle is more general: if the probability of acquiring resistance is low, rare resistance should be generated by hard selective sweeps.

It is notable that we can detect differences in relative changes in diversity, given the constraints of our data. Our data are cross-sectional, from different patients treated with different regimens from many studies over more than two decades. We have sequences from only one or two genes, and so it is possible that there are DRMs or other selected mutations driving population dynamics outside of the sequenced regions (particularly among patients treated with protease inhibitors). The DRMs that are observed may be associated with variable selection coefficients and may therefore differ in their rate of expansion in the population and thus the associated signature of the selective sweep. Additionally, DRMs occurring later in the course of an infection may have smaller beneficial effects as compared to early DRMs. Different patients may have varying profiles of diversity at the time of diagnosis, which could also effect whether sweeps will be hard or soft. All of these items could serve to obscure the relationship between DRM fixation and changes in diversity, but our results appear robust despite the substantial noise.

Although our results appear robust, certain systematic biases could also affect our results. First, we observe an increase in the number of ambiguous calls through our study period. This is likely due to a greater awareness of within-patient diversity and improvements in sequencing technology. Having more ambiguous reads called in later years gives us more power to detect large decreases in diversity as compared to earlier years. As effective treatments come from later in the study period, this could lead to more observed hard sweeps among effective treatments. However, we believe that our conclusion is robust to this systematic bias as we used a conservative $p$-thinning procedure that explictly equalized the power to detect hard and soft sweeps across years.

Time can also factor into our analysis in other ways that are harder to control. Improved disease monitoring may mean that modern patients have a shorter time between the appearance of a DRM and its sequencing and discovery. This could result in hard selective sweep signatures that are more pronounced in later years because they are detected before they have had time to erode. However, Pennings et al. estimate that sweep signatures require a substantial amount of time to erode (on the order of months) (*Pennings et al., 2014*), so we do not believe that this effect alone can explain the pattern. Alternatively, if DRMs acquired in response to earlier treatments have lower selection coefficients than treatments given more recently, early selective sweeps would take longer. Recombination could move DRMs to multiple genetic backgrounds before the mutation fixes in the population. This would also also result in muted changes in diversity among early treatments that could not be

readily distinguished from the signatures of soft sweeps. However, there is no evidence that selection coefficients of DRMs among early treatments are smaller.

Finally, because our primary measure to understand the effect of the fixation of a drug resistance mutation is an internal comparison with other patients on the same treatment, we do not believe that our results could be generated solely by data heterogeneity. Further, that our results are robust to different measurements and filtering approaches suggests DRMs do indeed sweep differently in HIV populations subject to more or less effective treatments.

Drug resistance evolution is no longer the threat to HIV patients it once was and treatments exist that almost never lead to resistance. However, our observation that effective drug combinations lead to hard selective sweeps could be useful for improving treatments for other pathogens. Even among relatively small samples, if patients who acquire drug resistance mutations do not have a significant decrease in diversity relative to those who did not acquire drug resistance mutations, this might suggest that the fixing mutations are occurring via soft selective sweeps, and the treatment brings patients into a dangerous regime for drug resistance. Alternatively, if patients acquire drug resistance mutations, but those patients also have very little diversity, it may be that this is a safer treatment and that the emergence and fixation of drug resistance mutations was a relatively uncommon occurrence.

Looking at changes in diversity following a sweep in order to assess the mode of adaptation could be particularly well-suited to looking at evolution of cancer. While single cell sequences isolated from tumors have yielded promising insights about evolutionary dynamics, the process is invasive and relatively difficult. Sequencing tumor-free cancerous cells circulating in the blood provides less information, but can be done serially and provides a good measure of tumor heterogeneity. Applying a method such as ours to monitor changes in cell-free DNA diversity over time may allow us to determine if certain treatments reproducibly lead to soft sweeps and thus are very likely to fail in general.

Comparing treatment effectiveness with the occurrence of soft and hard selective sweeps may also provide supplementary information about additional risk factors among patients. In the case of HIV, we find that high effectiveness is associated with hard selective sweeps, which suggests that the the virus has a hard time evolving drug resistance, and the patients in whom resistance evolved are merely the unlucky ones. However, there may be cases where failure is rare, but associated with soft selective sweeps. Such a situation thus reflects a discordance between what happens within patients and what we see at a population level. This, in turn, may be indicative of a behavioral, genetic or virologic difference among the groups of patients and efforts should be made to find out how failing patients are different from non-failing patients.

In conclusion, we find that the study of diversity in viral populations with resistance can show differences in evolutionary pathways of adapting pathogenic populations and provides a concrete example of how population genetics theory can make substantive predictions about medically relevant problems. Next generation and single molecule sequencing have the capacity to bring much more precision in determining the dynamics of within-patient populations. However, we also urge researchers and clinicians to report more information concerning the diversity of pathogen populations, even in the form of minor allele frequency cut offs for calling ambiguous calls or raw sequencing data, as this might allow new insight from data that might be otherwise overlooked.

## Materials and methods

### Data collection and filtering

Direct PCR (D-PCR) dataset

We collected one consensus sequence of the HIV-1 reverse transcriptase gene per patient from 6717 patients across 120 different studies from the Stanford HIV Drug Resistance Database (*Rhee et al., 2003*). These patients represent all individuals in the database with HIV populations which were treated with exactly one drug regimen and that had an associated reverse transcriptase sequence. Protease sequences were also recorded, when available (5163 sequences).

Nearly all patients received treatment that included one or more nucleoside reverse transcriptase inhibitors (NRTI). In many patients, the NRTIs were paired with either a non-nucleoside reverse transcriptase inhibitor (NNRTI) or a protease inhibitor (PI), which was in some patients boosted with a

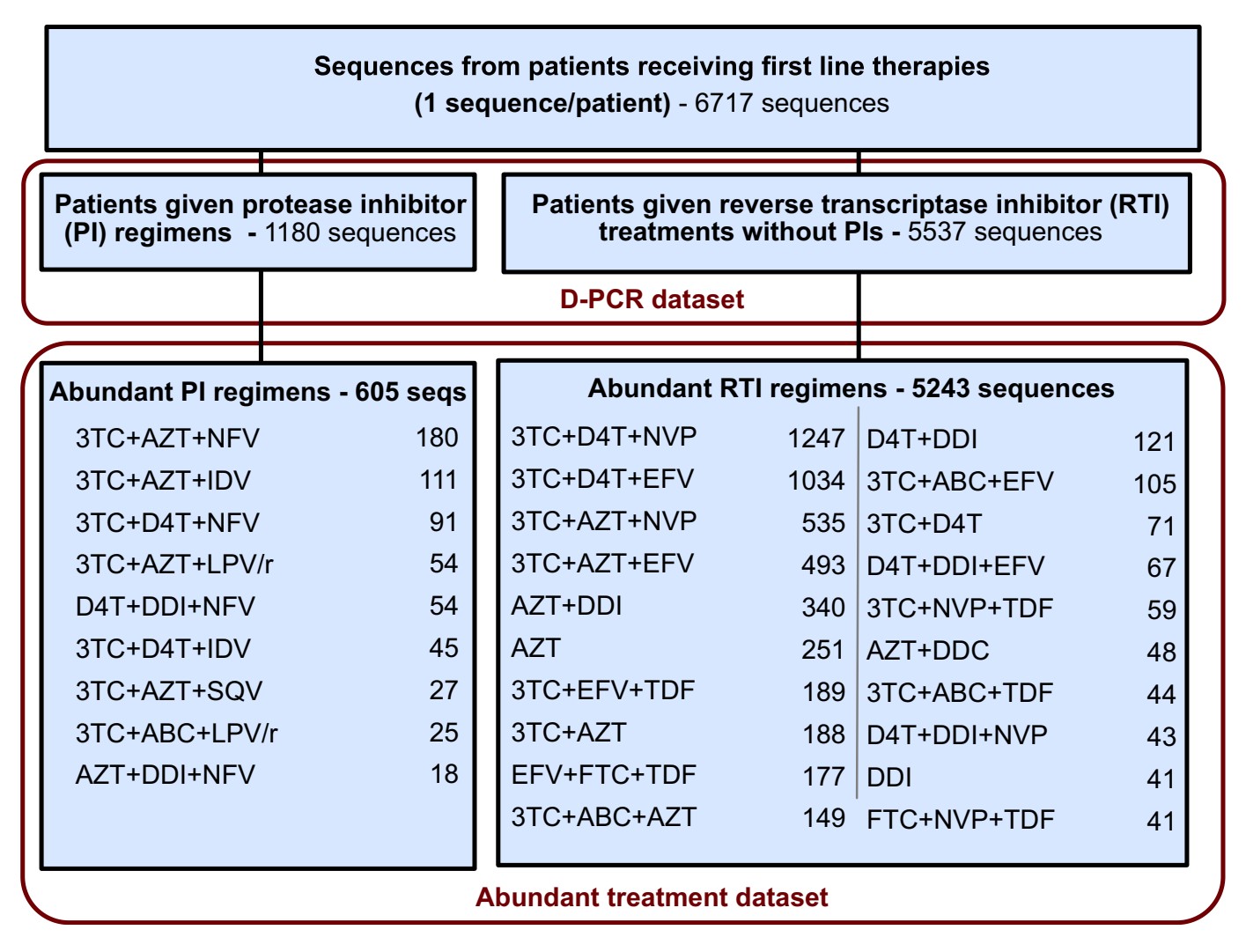

**Figure 5.** Summary of data subgroups and filtering. Summary of all sequences from patients receiving first line therapy used throughout the analysis. The dataset is broken down into patients receiving protease inhibitor (PI) therapy with reverse transcriptase inhibitors (RTIs) and patients receiving only RTIs. The abundant treatment dataset shows treatments given to many patients within our dataset. For full filtering parameters for the abundant dataset, see Materials and methods: Data collection & filtering. Counts of sequences for each treatment are given.

The following figure supplements are available for figure 5:

**Figure supplement 1.** Distribution of number of DRMs by treatment.

**Figure supplement 2.** Data is heterogenous with respect to year in terms of the number of drugs per treatment, the diversity of treatments and the number of ambiguous reads per sequence.

**Figure supplement 3.** *p*-thinning controls for systematic changes in the called number of ambiguous reads over time A.

**Figure supplement 4.** The number of ambiguous reads is approximately distributed according to a negative binomial when *p*-thinned relative to 1989.

**Figure supplement 5.** The number of ambiguous reads is approximately distributed according to a negative binomial when *p*-thinned relative to 1995.

low dose of ritonavir, a second PI, in order to boost drug levels and improve effectiveness. Among patients receiving PIs, only patients with both protease and reverse transcriptase sequences were included. All sequence information in the HIV database is recorded as aligned consensus nucleotide

sequences. The data are taken across many studies, each with their own procedures and cutoffs for calling positions ambiguous. No electropherogram data were available and the ambiguous calls reported were taken as submitted to the database. The sequences from these patients were labeled the direct PCR (D-PCR) dataset.

### Abundant treatment dataset

For each sequence we counted the number of relevant drug resistance mutations (DRMs, see below). For a portion of the analysis, we rely on examining many HIV-1 sequences from patients being treated with the same drug regimen. Because we measure the association between the number of DRMs and genetic diversity by examining a slope ($\Delta_{DRM}$), our signal could be heavily influenced by single patients if the distribution of patients with a certain number of DRMs within a given treatment is narrow (e.g., a treatment has almost only patients with 0 DRMs). To ensure that that our observed signal is not driven by these cases, we exclude treatments if among the sequences from a treatment we do not have at least three sequences within three of the DRM categories (0, 1, 2, 3 or 4 DRMs). This restriction yielded a dataset with 5848 sequences from 29 treatments and was termed the abundant treatment dataset. The breakdown of the D-PCR dataset and the abundant treatment dataset by the type of treatment a patient got (PI-based treatment and RTI-based treatment) is listed in *Figure 5*. Distributions of number of DRMs per sequence are shown in the supplement for all treatments (*Figure 5—figure supplement 1*).

### Clonal dataset

We supplemented our analysis with an additional dataset of patients from which multiple HIV-1 strains were sampled. This dataset comprised 10,235 sequences, but from a relatively small number of patients ($n$ = 174) with much less variety in their given treatments (the mean number of sequences per patient was 58 with an interquartile range of 16 to 64). These sequences were derived from taking a within-patient population of HIV-1 and attempting to isolate, clone and sequence single strains. We used these reads (termed the ''clonal dataset'') to validate ambiguous calls as a within-patient diversity measure.

## Sequence processing

Sequences from reverse transcriptase and protease were analyzed to determine the number of ambiguous calls and the number of drug resistance mutations per sequence.

We called a nucleotide non-ambiguous if it read A, T, C or G, and grouped lowercase (and less confident) a, t, c and g calls with their capital counterparts. Nucleotides called as W, S, M, K, R, Y (ambiguity between two nucleotides) and B, D, H, V (ambiguity between three nucleotides) and their lowercase counterparts were included as ambiguous calls. Ns and Xs (indicating no information about the identity of the position) were excluded.

We also examined ambiguities on the amino acid level by using nucleotide level information. If, for example, a nucleotide triplet was recorded as AAM, where M indicates an adenine/cytosine ambiguity, the amino acid at that position was ambiguous between AAA (Lysine, K) and AAC (Asparagine, N). The amino acid for that position would then be recorded K/N. All ambiguous calls at the nucleotide level were translated into ambiguous calls at the amino acid level, including if the ambiguous call reflected synonymous encodings (i.e., AAA and AAG are both Lysine, and the amino acid would be encoded K/K). The number of ambiguous amino acids was recorded for each sequence.

We determine whether mutations are associated with drug resistance based on the 2009 update of DRMs for the surveillance of transmitted HIV-1 drug resistance adopted by the World Health Organization (*Bennett et al., 2009*), which lists drug resistance mutations that are indicative of selective pressure. For each patient, we determine the number of mutations that confer drug resistance at the amino acid level to any of the classes of drugs the patient is receiving (i.e., NRTI, NNRTI, PI). These drug resistance mutations are counted as fixed if they are either non-ambiguously the resistant type, or they are an ambiguous call with all possible states as the resistant type (i.e., N/N in the example above).

5163 patients in the D-PCR dataset had a sequenced protease gene available in the database (77% of patients). Not all patients had entire reverse transcriptase and protease genes sequenced,

but only 1% of sequences had fewer than 500 basepairs sequenced of 1680 possible in reverse transcriptase and only 1% of available protease sequences had fewer than 210 of 300 bases in protease.

## Validation of ambiguous calls as genetic diversity measure

In order to validate the appropriateness of ambiguous calls as a proxy for genetic diversity, we computed diversity using an ambiguous call measure and compared it to a diversity measure that did not use the ambiguous call data. For each site, we computed genetic diversity using the ambiguous calls as the proportion of all D-PCR HIV-1 sequences that had an ambiguous basepair call at that site. This approximates the percentage of patients with within-patient diversity by site.

In order to test if this measure of diversity based on ambiguous calls correlates well with other measures of diversity that don't depend on ambiguous calls, we used the clonal dataset to compute the average sitewise $\pi$. For a site with nucleotides $A$, $T$, $C$ and $G$ at within-patient frequencies $p_A$, $p_T$, $p_C$ and $p_G$, $\pi$ is computed as

$$\pi = 1 - \sum_{i \in \{A,T,C,G\}} p_i^2$$

$\pi$ is equal to zero when every sequence has the same basepair call (i.e., all As) and is maximized when multiple categories are at intermediate frequencies (an even split between A, T, C and G). Within-patient diversity was measured by first computing $\pi$ at each site separately within each patient and then averaging over all patients. The $\pi$ calculation at a site for a particular patient was only included if the patient had at least two sequences calling non-N identity at that site. We computed diversity in the same way at the amino acid level to validate that our signals persisted when looking at codons.

## Clinical effectiveness of antiretroviral treatments

We expect that clinical effectiveness of an HIV treatment affects the probability of the virus undergoing a soft or hard selective sweep. All HIV treatment regimens that occurred at least 50 times in the D-PCR dataset were evaluated for treatment effectiveness based on a literature review. As a measure for treatment effectiveness, we recorded the proportion of patients whose treatment was still successful after a year of treatment, as indicated by a viral load of $\leq$ 50 copies of HIV-1 RNA/mL or less after 48 or 52 weeks in an on-treatment analysis. Our literature review was mostly based on the papers reviewed in Lee et al (*Lee et al., 2014*). Because this review did not include review information for several older treatment regimens, we supplemented our analysis with additional studies. A full description of how clinical treatment effectiveness was calculated by study can be found in the supplement (*Supplementary files 1* and *2*: Determining treatment effectiveness). A second researcher randomly chose 5 studies and independently followed the protocol to determine treatment effectiveness for these studies, providing confirmation of our method. Because we believe the thus collected information may be useful to other researchers, we provide our estimates in the supplement (*Supplementary file 3*).

## *p*-thinning to adjust for the effect of year

Because the drug regimens systematically changed over time (*Figure 5—figure supplement 2A*), it is essential to ensure that our estimates of diversity are not confounded by the changes in sequencing practices over the same time. While all samples were Sanger sequenced, we do find that the number of recorded ambiguous calls increased over time (*Figure 5—figure supplement 2C*), possibly because the cut-off of calling a read as ambiguous became lower. This effect, taken on aggregate across all sequences was not significant (p=0.09, linear regression with year predicting the number of ambiguous calls). However, when examining only sequences with 0 DRMs, a strong positive correlation emerges, with each year associated with 0.56 more ambiguous calls per sequence (p=4.9 $\times$ 10$^{-8}$). That the difference in effect can be seen in the zero DRM class but not among sequences with any number of DRMs underscores the strong interaction between low diversity and multiple DRMs among modern treatments.

Because the probability of calling a nucleotide as ambiguous has increased over time, we have potentially greater power to detect changes in the number of ambiguous reads with increasing number of DRMs. To account for this change in power, we used a p-thinning procedure (*Grandell, 1997*)

to control for the effect of year on the number of ambiguous calls. This allowed us to attribute differences in the relationship between the number of DRMs and diversity to treatment and not to increased probability of calling reads as ambiguous in later years.

We first measured the effect of year on diversity by fitting a linear model to the number of ambiguous calls in each year among all sequences with zero observed DRMs. By limiting our analysis to only sequences without DRMs, we remove the hypothesized change in diversity expected following a sweep. We observed the following relationship between year and the number of ambiguous calls:

$$\text{(Number of ambiguous reads)} = f(\text{Year}) = -505.92 + 0.26 * \text{Year} \tag{1}$$

According to this relationship, we computed probabilities of calling ambiguous reads in different years relative to the first year sampled (here, 1989). We subsampled our data using these probabilities so that late and early years had comparable numbers of ambiguous calls. For example, in 2000, nearly twice as many ambiguous calls were reported as compared to 1989. Therefore, for each ambiguous call observed in a sequence from 2000, we include it in our sample with probability ~1/2. Intuitively, this translates observations from later years into units of ambiguous calls that are comparable to those observed in 1989. The subsample effect for year $i$ with respect to 1989 ($p_{1989,i}$) is calculated as $P_{1989,i} = f(1989)/f(i)$ (see *Equation 1*). The fit of *Equation 1* to the year means and the thinning effect $P_i$ are shown in *Figure 5—figure supplement 3A* in black.

Because very few ambiguous calls were made in 1989, p-thinning all read counts to be comparable to sequences from 1989 lowers the resolution of our data. We therefore also performed subsampling using only data after and including 1995, which retained more ambiguous calls. This particular cutoff was chosen for two reasons. First, ambiguous calls were not reliably recorded until several landmark papers studying within-patient diversity were published in 1993 (*Larder et al., 1993*; *Piatak et al., 1993*). Second, before 1995, we have <20 sequences/year, so we have much lower resolution to determine the rates of ambiguous read calling. This excluded only 45 sequences, and changed the linear model fit only slightly (see *Equation 2* and *Figure 5—figure supplement 3A*).

$$\text{(Number of ambiguous reads)} = f_{1995}(\text{Year}) = -465.96 + 0.24 * \text{Year} \tag{2}$$

Using data only after and including 1995, the subsample effect for year $i$ ($P_{1995,i}$) is calculated as $P_{1995,i} = f_{1995}(1995)/f_{1995}(i)$. Although the linear fits are similar, the effect of the subsampling is less severe (see *Figure 5—figure supplement 3B*). This is because later years are rescaled to be comparable to 1995 observations as opposed to 1989 observations. Observations from 1995 have a greater number of ambiguous calls.

In including each ambiguous call with probability relative to the subsample effect, for each sequence $j$ sampled in year $i$, the number of ambiguous calls for sequence $j$ is distributed as Poisson( (Number of ambiguous calls on sequence $j$) $\times P_i$ ). Note, samples taken from the reference year (i.e., 1989 or 1995) were also re-drawn according to a Poisson distribution with $\lambda = 1$. This process is known as p-thinning (*Grandell, 1997*), and is similar to standard bootstrapping and $m$ out of $n$ resampling (see [*Politis et al., 1999*]). These subsampled counts for the number of ambiguous calls are used throughout the analysis below.

## Quantifying the relationship between clinical effectiveness and diversity reduction

We estimate the relationship between the number of DRMs and genetic diversity by fitting a generalized linear mixed model (GLMM) with a negative binomial error distribution for our 29 abundant treatments. We found that the subsampled data visually fit a negative binomial distribution much more closely than a Poisson distribution, which is often used for count data (See *Figure 5—figure supplement 4* and *Figure 5—figure supplement 5*). In this model, length, the number of DRMs and an intercept term are fit as fixed effects, and the number of DRMs by treatment is fit as a random effect. This allows us to assess the relationship between diversity and the number of ambiguous reads separately for each treatment. The models were fit using the glmmADMB package in R (*Fournier et al., 2012*).

$$\text{Subsampled number of ambiguous reads} \sim \Delta_{DRM,all}(\text{numDRM}) + \alpha_{all} + \gamma(\text{Sequence Length}) \\ + \ (\alpha_t + \Delta_{DRM,t}(\text{numDRM})|\text{Regimen}) \tag{3}$$

This model was fit to 1000 datasets created by p-thinning the number of ambiguous calls. The overall effect of a DRM on diversity was fit by the $\Delta_{DRM,all}$ term, but the effect of a DRM on diversity by treatment $t$ is fit by the random effect term, $\Delta_t$. The full effect of a DRM on diversity for a given treatment was called $\Delta_{DRM}$ and was computed by combining the treatment-specific random effect and the overall fixed effect of the model ($\Delta_{DRM} = \Delta_{DRM,t} + \Delta_{DRM,all}$). Confidence intervals were generated by excluding the highest and lowest 2.5% of estimates of $\Delta_{DRM}$ among the subsamples.

We performed the above procedure three times: our main analysis used the $p_{1995,i}$ procedure for p-thinning sequences from year $i$ and included sequences with 4 or fewer DRMs. We performed this same analysis using $p_{1995,i}$-thinning and including all numbers of DRMs and using $p_{1989,i}$-thinning and including only sequences with 4 or fewer DRMs.

To quantify the effect of treatment effectiveness on $\Delta_{DRM}$, we used both a parametric and a non-parametric approach.

## Parametric approach

To compare how the effect of DRMs on genetic diversity varied between two groups of treatments, we fit generalized linear models (GLMs) with a negative binomial error distribution in R (**R Core Team, 2013**) using the package pcsl (**Jackman, 2015**) including all sequences belonging to the 29 treatments that passed our threshold criteria (see above).

These models were parametrized to fit separate slopes for the four different types of treatments (1, 2 or 3 NRTIs, 2NRTIs + NNRTI, 2NRTIs + PI, 2NRTIs + PI/r). To do this, we used an indicator variable for membership in each of the four groups and fit the following model:

$$\begin{aligned} \text{Subsampled number} \\ \text{of ambiguous reads} \end{aligned} \sim \ \alpha + \gamma(\text{Sequence Length}) \\ + \ \text{numDRM} * [\Delta_{1,2,3NRTI}\ (\mathbb{1}_{1,2,3NRTI}) + \Delta_{2NRTI+NNRTI}\ (\mathbb{1}_{2NRTI+NNRTI}) \\ + \ \Delta_{2NRTI+PI}(\mathbb{1}_{2NRTI+PI}) + \Delta_{2NRTI+PI/r}(\mathbb{1}_{2NRTI+PI/r})] \tag{4}$$

The four $\Delta$ terms measure the change in diversity associated with the acquisition of an additional DRM separately among the four treatment types. This model was refit 1000 times to different datasets generated through p-thinning. We compared the effect of a DRM on diversity between more and less effective treatments by performing Wilcoxon rank-sign tests between $\Delta$ coefficient fits to the 1000 subsamples ($\Delta_{1,2,3NRTI}$ versus $\Delta_{2NRTI+NNRTI}$ and $\Delta_{2NRTI+PI}$ versus $\Delta_{2NRTI+PI/r}$).

## Non-parametric approach

Apart from discriminating between two treatment categories, we also tested the association between our continuous measure of treatment effectiveness and the change in diversity associated with a DRM on a given treatment ($\Delta_{DRM}$) as a non-parametric approach. We then fit a linear regression between treatment effectiveness and the corresponding $\Delta_{DRM}$ from the GLMM values to quantify the effect of treatment effectiveness on the diversity reduction associated with drug resistance mutations. For this analysis, we excluded the treatment 3TC+D4T+LPV/r in our GLMM, because, although we know its effectiveness information, it did not pass the inclusion criteria of having enough sequences with different numbers of DRMs (see description of the abundant treatement dataset). To measure the overall relationship, we fit a linear regression predicting $\Delta_{DRM}$ for each treatment across 1000 subsampled runs using corresponding treatment effectiveness. We plot the median of our regressions and our confidence interval is plotted using the central 95% of regressions.

## Acknowledgements

The authors thank Alison Hill, Daniel Rosenbloom, Heather Machado, Nandita Garud, Ben Wilson, Emily Ebel, Zoe Assaf, Rajiv McCoy, Richard Neher, Joachim Hermisson and members of the Petrov lab for comments and discussion. SYR, SPH and RWS were supported in part by NIH grant R01 AI068581. The work was partially supported by the grants R01GM100366 and R01GM097415 to

DAP. This material is based upon work supported by the National Science Foundation Graduate Research Fellowship to AFF under Grant No. DGE-114747.

## Additional information

### Funding

| Funder | Grant reference number | Author |
| --- | --- | --- |
| National Institutes of Health | R01 AI068581 | Soo-Yon Rhee<br>Susan P Holmes<br>Robert W Shafer |
| National Science Foundation | NSF GRFP, DGE-114747 | Alison F Feder |
| National Institutes of Health | R01 GM100366 | Dmitri A Petrov |
| National Institutes of Health | R01 GM097415 | Dmitri A Petrov |

The funders had no role in study design, data collection and interpretation, or the decision to submit the work for publication.

### Author contributions

AFF, DAP, PSP, Conception and design, Analysis and interpretation of data, Drafting or revising the article; S-YR, RWS, Acquisition of data, Analysis and interpretation of data, Drafting or revising the article; SPH, Analysis and interpretation of data, Drafting or revising the article

## Additional files

### Supplementary files

• Supplementary file 1. Detailed description of treatment effectiveness computation including references. (A) Detailed description (organized by study) of how we extracted treatment successes versus failures, as well as length of study and viral load limit for each study. Part (B) is a reorganization of (A), but it excludes specific details on how we calculated each number.

• Supplementary file 2. Detailed description of treatment effectiveness computation including references. (B) Chart summary (organized alphabetically by treatment) detailing all studies considered for our treatment effectiveness analysis. For each study, we recorded the number of weeks, the virologic load limit under which was considered a ''success,'' the number of successes and the total number of patients on on-treatment analysis considered.

• Supplementary file 3. Part (C) is a further summary of the final treatment effectiveness used throughout our analysis. This supplement has a full reference section describing the studies used.

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
