## [Decision Letter]

Thank you for submitting your work entitled "More efficacious drugs lead to harder selective sweeps in the evolution of drug resistance in HIV-1" for consideration by *eLife*. Your article has been reviewed by two peer reviewers, and the evaluation has been overseen by a Reviewing Editor and Aviv Regev as the Senior Editor.

The reviewers have discussed the reviews with one another and the Reviewing editor has drafted this decision to help you prepare a revised submission.

Summary:

In this paper, it is shown that the presence of drug-resistant mutations result in lower in-patient diversity of the virus, suggesting that sweeps are occurring. It is also shown that more effective HIV treatments (according to several related definitions) lead to an increased anticorrelation between diversity and the number of drug resistance mutations (DRMs). This is interpreted as evidence that more efficacious drugs lead to harder selective sweeps and hence greater reductions in diversity per DRM. Overall this is an interesting manuscript that does its best to use data collected for another purpose to study an interesting issue in evolutionary dynamics.

Specific points:

The following points need to be addressed, however, before the paper can be considered for publication in *eLife*.

1) The Introduction would benefit from emphasizing the following points more carefully. The motivation for distinguishing between soft and hard selective sweeps is not described adequately (the discussion contains most of the salient points). It is also important to note in the very first paragraph that this paper is restricted to drug resistance mutations in treated populations. In untreated populations, the story is very different. The fact that more efficacious treatments lead to fewer resistant mutations is almost tautological. The connection between this effectiveness and a signature of the evolutionary dynamics of resistance is nice, and that is the point that needs to be emphasized.

2) Ambiguous calls from Sanger sequencing are used as a proxy for diversity because this is the best that can be done given the available data. A convincing argument is made suggesting that the ambiguous calls are at least a reasonable proxy, based on earlier work and analysis of data from multiple clones sequenced from individual patients. However, there could be additional factors correlated with treatment efficacy that also affect the fraction of ambiguous calls. For example, there may be an effect of the timing of the sample relative to the progression of the disease (e.g. relative to treatment failure). Since there is only one sample per individual, it is easy to imagine that sampling is done more often at treatment failure for less efficacious drugs, but not for more efficacious drugs. This doesn't necessarily confound the analysis, but it could potentially lead to a correlation between efficacy and ambiguous calls that does not have to do with the hardness of the sweeps. These sorts of caveats are unavoidable in this kind of study, but the potential problems should be acknowledged.

3) In Figure 3, why is the y-intercept higher for more efficacious drugs (i.e. why is diversity higher for samples with 0 DRMs)? It seems that the opposite could be true, since the more efficacious drugs should maintain the virus at lower population sizes, leading to less diversity.

---

## [Author Response]

*The following points need to be addressed, however, before the paper can be considered for publication in* eLife

*. 1) The Introduction would benefit from emphasizing the following points more carefully. The motivation for distinguishing between soft and hard selective sweeps is not described adequately (the discussion contains most of the salient points). It is also important to note in the very first paragraph that this paper is restricted to drug resistance mutations in treated populations. In untreated populations, the story is very different. The fact that more efficacious treatments lead to fewer resistant mutations is almost tautological. The connection between this effectiveness and a signature of the evolutionary dynamics of resistance is nice, and that is the point that needs to be emphasized.*

We thank the reviewers for noting ways in which the most important points of the paper can be better highlighted and motivated. To this end, we have expanded the discussion of several things that were unclear in the Introduction, and most substantively, expanded the discussion concerning the importance of connecting evolutionary dynamics to the probability of eventual failure (Introduction, paragraphs 4/5).

*2) Ambiguous calls from Sanger sequencing are used as a proxy for diversity because this is the best that can be done given the available data. A convincing argument is made suggesting that the ambiguous calls are at least a reasonable proxy, based on earlier work and analysis of data from multiple clones sequenced from individual patients. However, there could be additional factors correlated with treatment efficacy that also affect the fraction of ambiguous calls. For example, there may be an effect of the timing of the sample relative to the progression of the disease (e.g. relative to treatment failure). Since there is only one sample per individual, it is easy to imagine that sampling is done more often at treatment failure for less efficacious drugs, but not for more efficacious drugs. This doesn't necessarily confound the analysis, but it could potentially lead to a correlation between efficacy and ambiguous calls that does not have to do with the hardness of the sweeps. These sorts of caveats are unavoidable in this kind of study, but the potential problems should be acknowledged.*

We appreciate that we are working with a very noisy dataset with many potential confounding factors and it is important to be direct about the ways in which they might influence our results. While much of the information concerning caution about our analysis was present in the original draft, we have consolidated the information into an expanded set of paragraphs in the Discussion. We have further grouped these caveats into separate categories: 1) data-specific noise factors that should obscure the signal, 2) properties of the data that might contribute to our observed signal that we control for within the analysis and 3) properties of the data that might contribute to our observed signal that are not possible to control. For each of these potential biases, we discuss why we do not believe that they generate the patterns seen in our data. The particular issue related to treatment monitoring described by the reviewers is included in this third category. We believe that this new format should make it more apparent to readers the types of factors that must be considered in interpreting our results.

*3) In Figure 3, why is the y-intercept higher for more efficacious drugs (i.e. why is diversity higher for samples with 0 DRMs)? It seems that the opposite could be true, since the more efficacious drugs should maintain the virus at lower population sizes, leading to less diversity.*

In investigating this issue, we found that the y-intercept was indeed higher for these more efficacious treatments. This effect is caused by the positive relationship between year and the probability of calling ambiguous reads. Therefore, better treatments (from later years) have a higher 0 DRM category than worse treatments (from earlier years). We controlled for the effect of year in the model, but the particular way in which that graph was plotted did not (and cannot) properly display the effect of year across the different treatments (as each treatment is comprised of patients from across multiple years). The more parsimonious solution to display this effect would be to rescale the number of ambiguous reads so that they are comparable across years.

We therefore modified our analysis slightly in order to account for this effect through employing a *p*-thinning procedure, in which the number of ambiguous reads per sequence was subsampled so that year effect was controlled for explicitly. Switching to this approach did not change any of our results in a substantive manner, and offered several advantages over our original approach:

A) We now examine ambiguous read data as counts instead of proportions, which better preserves the variance structure inherent in our data;

B) Since the number of ambiguous calls does not increase over time, we have comparable power to detect changes in the number of ambiguous calls with different numbers of DRMs among early treatments and late treatments;

C) Confidence intervals and summary statistics can be constructed from the different datasets of subsampled ambiguous reads, which allows for the results to be more interpretable.

To ensure that this approach was statistically sound, we had extensive conversation with Susan P Holmes of the Department of Statistics at Stanford University. We further believe that her contributions to improving the work merits authorship, and the author list has been updated accordingly.

In our new updated version of Figure 3 (and their corresponding supplemental figures), we now plot model fits separately for the four different treatment categories, as opposed to the specific slopes for each treatment, which, upon further reflection, we believe were adequately captured in Figure 3. This new figure shows how the intercepts within the model are set to have the same 0 DRM diversity. This can now be done because the diversity for treatment types from different years are rescaled to be comparable before the model is fit.

The majority of the updated text in our resubmitted manuscript describes the *p*-thinning process and updated analysis in full detail. Although *p*-thinning makes the manuscript slightly more complicated, we believe that this approach is the best way to account for year changing throughout our sample period, and ultimately the fairest way to compare early and late treatments.